# DAPO: An Open-Source LLM Reinforcement Learning System at Scale

Qiying Yu[1,2,4]*   Zheng Zhang[1]*   Ruofei Zhu[1]   Yufeng Yuan[1]   Xiaochen Zuo[1]   Yu Yue[1]
Weinan Dai[1,2,4]‡   Tiantian Fan[1]‡   Gaohong Liu[1]‡   Juncai Liu[1]‡   Lingjun Liu[1]‡   Xin Liu[1]‡
Haibin Lin[1]‡   Zhiqi Lin[1]‡   Bole Ma[1]‡   Guangming Sheng[1,3]‡   Yuxuan Tong[1,2,4]‡   Chi Zhang[1]‡
Mofan Zhang[1]‡   Ru Zhang[1]‡   Wang Zhang[1]‡   Hang Zhu[1]   Jinhua Zhu[1]   Jiaze Chen[1]
Jiangjie Chen[1,4]   Chengyi Wang[1]   Hongli Yu[1,2,4]   Yuxuan Song[1,2,4]   Xiangpeng Wei[1]
Hao Zhou[2,4]†   Jingjing Liu[2,4]   Wei-Ying Ma[2,4]   Ya-Qin Zhang[2,4]
Lin Yan[1,4]   Yonghui Wu[1]   Mingxuan Wang[1,4]†

* Equal contribution. ‡ Equal engineering contribution.

[1] ByteDance Seed   [2] Institute for AI Industry Research (AIR), Tsinghua University
[3] The University of Hong Kong
[4] SIA-Lab of Tsinghua AIR and ByteDance Seed

## Abstract

Inference scaling empowers LLMs with unprecedented reasoning ability, with reinforcement learning as the core technique to elicit complex reasoning. However, key technical details of state-of-the-art reasoning LLMs are concealed (such as in OpenAI o1 blog and DeepSeek R1 technical report), thus the community still struggles to reproduce their RL training results. We propose the **D**ecoupled Clip and **D**ynamic s**A**mpling **P**olicy **O**ptimization (**DAPO**) algorithm, and fully open-source a state-of-the-art large-scale RL system that achieves 50 points on AIME 2024 using Qwen2.5-32B base model. Unlike previous works that withhold training details, we introduce four key techniques of our algorithm that make large-scale LLM RL a success. In addition, we open-source our training code, which is built on the verl framework, along with a carefully curated and processed dataset. These components of our open-source system enhance reproducibility and support future research in large-scale LLM RL.

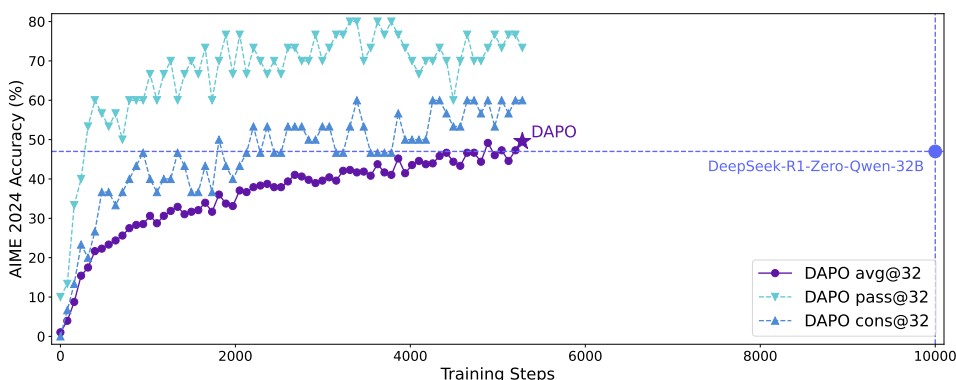

Figure 1: AIME 2024 scores of **DAPO** on the Qwen2.5-32B base model, outperforming the previous SoTA DeepSeek-R1-Zero-Qwen-32B using 50% training steps. The x-axis represents the gradient update steps.

† Correspondence to *zhouhao@air.tsinghua.edu.cn, wangmingxuan.89@bytedance.com*

39th Conference on Neural Information Processing Systems (NeurIPS 2025).

# 1 Introduction

Test-time scaling such as OpenAI's o1 [1] and DeepSeek's R1 [2] brings a profound paradigm shift to Large Language Models (LLMs) [3, 4, 5, 6, 7]. Test-time scaling enables longer Chain-of-Thought thinking and induces sophisticated reasoning behaviors, which makes the models superior in competitive math and coding tasks like AIME and Codeforces.

The central technique driving the revolution is large-scale Reinforcement Learning (RL), which elicits complex reasoning behaviors such as self-verification and iterative refinement. However, the actual algorithm and key recipe for scalable RL training remains a myth, hidden from technical reports of existing reasoning models [1, 2, 8, 9, 10, 11]. In this paper, we reveal significant obstacles in large-scale RL training and open-source a scalable RL system with fully open-sourced algorithm, training code and dataset that provides democratized solutions with industry-level RL results.

We experiment over Qwen2.5-32B [12] as the pretrained model for RL. In our initial GRPO run, we achieved only 30 points on AIME — a performance significantly below DeepSeek's RL (47 points). A thorough analysis reveals that the naive GRPO baseline suffers from several key issues such as entropy collapse, reward noise, and training instability. The broader community has encountered similar challenges in reproducing DeepSeek's results [13, 14, 15, 16, 17, 18, 19] suggesting that critical training details may have been omitted in the R1 paper that are required to develop an industry-level, large-scale, and reproducible RL system.

To close this gap, we release an open-source state-of-the-art system for large-scale LLM RL, which achieves 50 points on AIME 2024 based on Qwen2.5-32B model, outperforming previous state-of-the-art results achieved by DeepSeek-R1-Zero-Qwen-32B [2] (47 points) using 50% training steps (Figure 1). We propose the **D**ecoupled Clip and **D**ynamic s**A**mpling **P**olicy **O**ptimization (**DAPO**) algorithm, and introduce 4 key techniques to make RL shine in the long-CoT RL scenario. Details are presented in Section 3.

1. **Clip-Higher**, which promotes the diversity of the system and avoids entropy collapse;
2. **Dynamic Sampling**, which improves training efficiency and stability;
3. **Token-Level Policy Gradient Loss**, which is critical in long-CoT RL scenarios;
4. **Overlong Reward Shaping**, which reduces reward noise and stabilizes training.

Our implementation is based on verl [20]. By fully releasing our state-of-the-art RL system including training code and data, we aim to reveal valuable insights to large-scale LLM RL that benefit the larger community.

# 2 Preliminary

## 2.1 Proximal Policy Optimization (PPO)

PPO [21] introduces a clipped surrogate objective for policy optimization. By constraining the policy updates within a proximal region of the previous policy using clip, PPO stabilizes training and improves sample efficiency. Specifically, PPO updates the policy by maximizing the following objective:

$$\mathcal{J}_{\text{PPO}}(\theta) = \mathbb{E}_{(q,a)\sim\mathcal{D}, o_{\leq t}\sim\pi_{\theta_{\text{old}}}(\cdot|q)}$$
$$\left[ \min\left( \frac{\pi_\theta(o_t \mid q, o_{<t})}{\pi_{\theta_{\text{old}}}(o_t \mid q, o_{<t})} \hat{A}_t, \ \text{clip}\left( \frac{\pi_\theta(o_t \mid q, o_{<t})}{\pi_{\theta_{\text{old}}}(o_t \mid q, o_{<t})}, 1-\varepsilon, 1+\varepsilon \right) \hat{A}_t \right) \right], \tag{1}$$

where $(q, a)$ is a question-answer pair from the data distribution $\mathcal{D}$, $\varepsilon$ is the clipping range of importance sampling ratio, and $\hat{A}_t$ is an estimator of the advantage at time step $t$. Given the value function $V$ and the reward function $R$, $\hat{A}_t$ is computed using the Generalized Advantage Estimation (GAE) [22]:

$$\hat{A}_t^{\text{GAE}(\gamma,\lambda)} = \sum_{l=0}^{\infty} (\gamma\lambda)^l \delta_{t+l}, \tag{2}$$

where

$$\delta_l = R_l + \gamma V(s_{l+1}) - V(s_l), \quad 0 \leq \gamma, \lambda \leq 1. \tag{3}$$

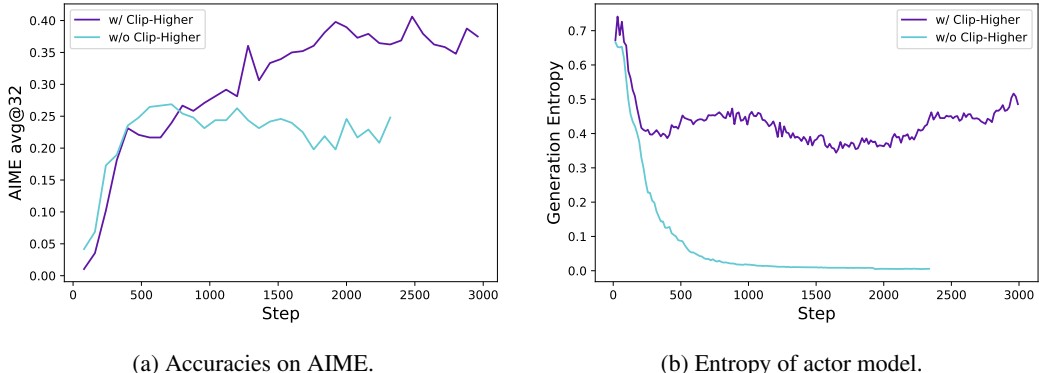

(a) Accuracies on AIME.

(b) Entropy of actor model.

Figure 2: The accuracy on the AIME test set and the entropy of the actor model's generated probabilities during the RL training process, both before and after applying **Clip-Higher** strategy.

## 2.2 Group Relative Policy Optimization (GRPO)

Compared to PPO, GRPO eliminates the value function and estimates the advantage in a group-relative manner. For a specific question-answer pair $(q, a)$, the behavior policy $\pi_{\theta_{\text{old}}}$ samples a group of $G$ individual responses $\{o_i\}_{i=1}^{G}$. Then, the advantage of the $i$-th response is calculated by normalizing the group-level rewards $\{R_i\}_{i=1}^{G}$:

$$\hat{A}_{i,t} = \frac{r_i - \text{mean}(\{R_i\}_{i=1}^{G})}{\text{std}(\{R_i\}_{i=1}^{G})}. \tag{4}$$

Similar to PPO, GRPO adopts a clipped objective, together with a directly imposed KL penalty term:

$$\mathcal{J}_{\text{GRPO}}(\theta) = \mathbb{E}_{(q,a) \sim \mathcal{D}, \{o_i\}_{i=1}^{G} \sim \pi_{\theta_{\text{old}}}(\cdot|q)}$$
$$\left[ \frac{1}{G} \sum_{i=1}^{G} \frac{1}{|o_i|} \sum_{t=1}^{|o_i|} \left( \min \left( r_{i,t}(\theta) \hat{A}_{i,t}, \ \text{clip}\left( r_{i,t}(\theta), 1 - \varepsilon, 1 + \varepsilon \right) \hat{A}_{i,t} \right) - \beta D_{\text{KL}}(\pi_\theta || \pi_{\text{ref}}) \right) \right], \tag{5}$$

where

$$r_{i,t}(\theta) = \frac{\pi_\theta(o_{i,t} \mid q, o_{i,<t})}{\pi_{\theta_{\text{old}}}(o_{i,t} \mid q, o_{i,<t})}. \tag{6}$$

It is also worth noting that GRPO computes the objective at the sample-level. To be exact, GRPO first calculates the mean loss within each generated sequence, before averaging the loss of different samples. As we will be discussing in Section 3.3, such difference may have an impact on the performance of the algorithm.

## 2.3 Removing KL divergence

The KL penalty term is used to regulate the divergence between the online policy and the frozen reference policy. In the RLHF scenario [23], the goal of RL is to align the model behavior without diverging too far from the initial model. However, during training the long-CoT reasoning model, the model distribution can diverge significantly from the initial model, thus this restriction is not necessary. Therefore, we will exclude the KL term from our proposed algorithm.

## 2.4 Rule-based reward modeling

The use of reward model usually suffers from the reward hacking problem [24, 25, 26, 27, 28, 29]. Instead, we directly use the final accuracy of a verifiable task as the outcome reward:

$$R(\hat{y}, y) = \begin{cases} 1, & \texttt{is\_equivalent}(\hat{y}, y) \\ -1, & \text{otherwise} \end{cases} \tag{7}$$

where $y$ is the ground-truth answer and $\hat{y}$ is the predicted answer. This is proved to be an effective approach to activating the base model's reasoning capability, as shown in multiple domains such as automated theorem proving [30, 31, 32, 33], computer programming [34, 35, 36, 37], and mathematics competition [2].

## 3 DAPO

We propose the **D**ecouple Clip and **D**ynamic s**A**mpling **P**olicy **O**ptimization (**DAPO**) algorithm. **DAPO** samples a group of outputs $\{o_i\}_{i=1}^{G}$ for each question $q$ paired with the answer $a$, and optimizes the policy via the following objective:

$$\mathcal{J}_{\text{DAPO}}(\theta) = \mathbb{E}_{(q,a)\sim\mathcal{D}, \{o_i\}_{i=1}^{G}\sim\pi_{\theta_{\text{old}}}(\cdot|q)}$$

$$\left[ \frac{1}{\sum_{i=1}^{G}|o_i|} \sum_{i=1}^{G}\sum_{t=1}^{|o_i|} \min\left( r_{i,t}(\theta)\hat{A}_{i,t}, \text{clip}\left( r_{i,t}(\theta), 1 - \varepsilon_{\text{low}}, 1 + \varepsilon_{\text{high}} \right)\hat{A}_{i,t} \right) \right] \quad (8)$$

$$\text{s.t.} \quad 0 < \left| \{o_i \mid \texttt{is\_equivalent}(a, o_i)\} \right| < G,$$

where

$$r_{i,t}(\theta) = \frac{\pi_\theta(o_{i,t} \mid q, o_{i,<t})}{\pi_{\theta_{\text{old}}}(o_{i,t} \mid q, o_{i,<t})}, \quad \hat{A}_{i,t} = \frac{R_i - \text{mean}(\{R_i\}_{i=1}^{G})}{\text{std}(\{R_i\}_{i=1}^{G})}. \quad (9)$$

The full algorithm can be found in Algorithm 1. In this section, we will introduce the key techniques associated with **DAPO**.

### 3.1 Raise the ceiling: Clip-Higher

In our initial experiments using naive PPO [21] or GRPO [38], we observed the entropy collapse phenomenon: the entropy of the policy decreases quickly as training progresses (Figure 2b). The sampled responses of certain groups tend to be nearly identical. This indicates limited exploration and early deterministic policy, which can hinder the scaling process.

We propose the **Clip-Higher** strategy to address this issue. Clipping over the importance sampling ratio is introduced in Clipped Proximal Policy Optimization (PPO-Clip) [21] to restrict the trust region and enhance the stability of RL. We identify that the upper clip can restrict the exploration of the policy, where making an 'exploitation' token more probable is much easier yet the probability of an unlikely 'exploration' token is too tightly bounded to be uplifted.

Concretely, when $\varepsilon = 0.2$ (the default value of most algorithms) and $\hat{A}_{i,t} > 0$ (the system tries to increase the probability), consider two actions with probabilities $\pi_{\theta_{\text{old}}}(o_i \mid q) = 0.01$ and $0.9$. The upper bounds of the increased probabilities $\pi_\theta(o_i \mid q)$ are $0.012$ and $1.08$, respectively ($\pi_{\theta_{\text{old}}} \cdot (1 + \epsilon)$). This implies that 'exploitation' tokens with a higher probability (*e.g.*, $0.9$) are not constrained to get even extremely larger probabilities like $0.999$. Conversely, for low-probability 'exploration' tokens, achieving a non-trivial increase in probability is considerably more challenging. Empirically, we also observe that the mean probability of up-clipped tokens is low: $\pi_\theta(o_i \mid q) < 0.2$ (Figure 3a). This supports our intuition that the upper clipping threshold indeed restricts the probability increase of low-probability 'exploration' tokens, thereby potentially constraining the exploration of the system.

Adhering to the **Clip-Higher** strategy, we decouple the lower and higher clipping range as $\varepsilon_{\text{low}}$ and $\varepsilon_{\text{high}}$, as highlighted in Equation 10:

$$\mathcal{J}_{\text{DAPO}}(\theta) = \mathbb{E}_{(q,a)\sim\mathcal{D}, \{o_i\}_{i=1}^{G}\sim\pi_{\theta_{\text{old}}}(\cdot|q)}$$

$$\left[ \frac{1}{\sum_{i=1}^{G}|o_i|} \sum_{i=1}^{G}\sum_{t=1}^{|o_i|} \min\left( r_{i,t}(\theta)\hat{A}_{i,t}, \text{clip}\left( r_{i,t}(\theta), 1 - \textcolor{red}{\varepsilon_{\text{low}}}, 1 + \textcolor{red}{\varepsilon_{\text{high}}} \right)\hat{A}_{i,t} \right) \right] \quad (10)$$

$$\text{s.t.} \quad 0 < \left| \{o_i \mid \texttt{is\_equivalent}(a, o_i)\} \right| < G.$$

We increase the value of $\varepsilon_{\text{high}}$ to leave more room for the increase of low-probability tokens. As shown in Figure 2, this adjustment effectively enhances the policy's entropy and facilitates the generation of more diverse samples. We keep $\varepsilon_{\text{low}}$ as it is, because increasing it will suppress the probability of these tokens to 0, resulting in the collapse of the sampling space.

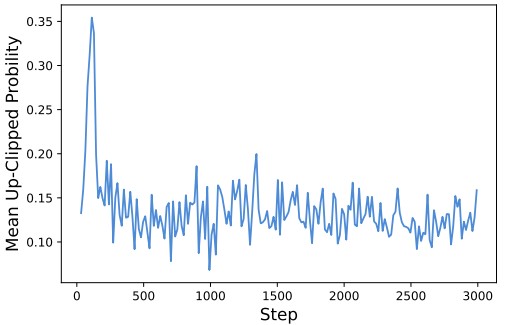 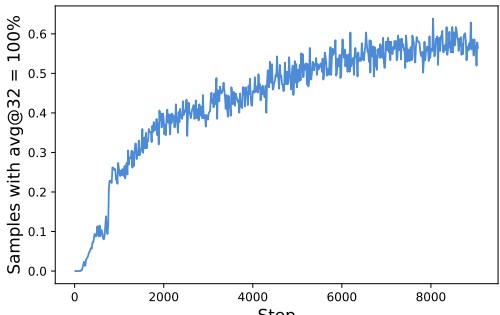

(a) Mean up-clipped probability.  (b) The proportion of samples with an accuracy of 1.

Figure 3: The mean up-clipped probability as well as the ratio of prompts with accuracy=1.

## 3.2 The more the merrier: Dynamic Sampling

Existing RL algorithm suffers from the gradient-decreasing problem when some prompts have accuracy equal to 1. For example for GRPO, if all outputs $\{o_i\}_{i=1}^G$ of a particular prompt are correct and receive the same reward, the resulting advantage for this group is *zero*. A zero advantage results in zero policy gradients, shrinking the magnitude and increasing the noise sensitivity of the batch gradient, thereby degrading sample efficiency. Empirically, the number of samples with accuracy equal to 1 continues to increase, as shown in Figure 3b. This means that the effective number of prompts in each batch keeps decreasing, which can lead to larger variance in gradient and dampens the gradient signals for model training.

To this end, we propose to **over-sample and filter out prompts with the accuracy equal to 1 and 0** as illustrated in Equation 11, leaving all prompts in the batch with effective gradients and keeping a consistent number of prompts. Before training, we keep sampling until the batch is fully filled with samples whose accuracy is neither 0 nor 1.

$$
\mathcal{J}_{\text{DAPO}}(\theta) = \mathbb{E}_{(q,a)\sim\mathcal{D},\{o_i\}_{i=1}^G\sim\pi_{\theta_{\text{old}}}(\cdot|q)}
$$
$$
\left[ \frac{1}{\sum_{i=1}^G |o_i|} \sum_{i=1}^G \sum_{t=1}^{|o_i|} \min\left( r_{i,t}(\theta)\hat{A}_{i,t},\ \text{clip}\left( r_{i,t}(\theta), 1 - \varepsilon_{\text{low}}, 1 + \varepsilon_{\text{high}} \right)\hat{A}_{i,t} \right) \right] \quad (11)
$$
$$
\text{s.t.} \quad 0 < \left| \left\{ o_i \mid \texttt{is\_equivalent}(a, o_i) \right\} \right| < G.
$$

Note that this strategy does not necessarily impede training efficiency, because the generation time is typically dominated by the generation of long-tail samples if the RL system is synchronized and the generation stage is not pipelined. Besides, we find that with dynamic sampling the experiment achieves the same performance faster as shown in Figure 6.

## 3.3 Rebalancing act: Token-Level Policy Gradient Loss

The original GRPO algorithm employs a sample-level loss calculation, which involves first averaging the losses by token within each sample and then aggregating the losses across samples. In this approach, each sample is assigned an equal weight in the final loss computation. However, we find that this method of loss reduction introduces several challenges in the context of long-CoT RL.

Since all samples are assigned the same weight in the loss calculation, tokens within longer responses may have a disproportionately lower contribution to the overall loss, which can lead to two adverse effects. First, for high-quality long samples, this effect can impede the model's ability to learn reasoning-relevant patterns within them. Second, we observe that excessively long samples often exhibit low-quality patterns such as gibberish and repetitive words. Thus, sample-level loss calculation, due to its inability to effectively penalize those undesirable patterns in long samples, leads to an unhealthy increase in entropy and response length, as shown in Figure 4a and Figure 4b.

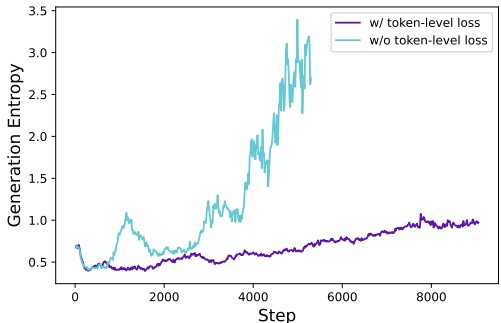 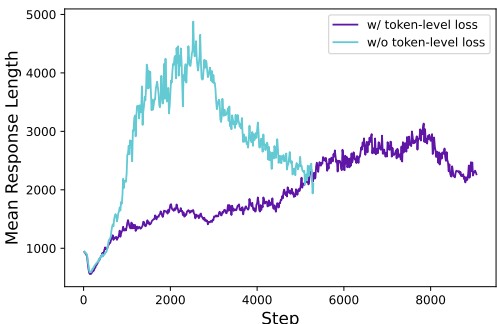

(a) Entropy of actor model's generation probabilities.    (b) Average length of actor model-generated responses

Figure 4: The entropy of the probability distribution of the actor model, as well as the changes in response length.

We introduce a **Token-level Policy Gradient Loss** in the long-CoT RL scenario to address the above limitations:

$$\mathcal{J}_{\text{DAPO}}(\theta) = \mathbb{E}_{(q,a)\sim\mathcal{D},\{o_i\}_{i=1}^{G}\sim\pi_{\theta_{\text{old}}}(\cdot|q)}$$

$$\left[ \frac{1}{\sum_{i=1}^{G}|o_i|}\sum_{i=1}^{G}\sum_{t=1}^{|o_i|}\min\left(r_{i,t}(\theta)\hat{A}_{i,t},\ \text{clip}\left(r_{i,t}(\theta), 1-\varepsilon_{\text{low}}, 1+\varepsilon_{\text{high}}\right)\hat{A}_{i,t}\right)\right], \quad (12)$$

$$\text{s.t.} \quad 0 < \left|\left\{o_i \mid \texttt{is\_equivalent}(a, o_i)\right\}\right| < G.$$

In this setting, longer sequences can have more influence on the overall gradient update compared to shorter sequences. Moreover, from the perspective of individual tokens, if a particular generation pattern can lead to an increase or decrease in reward, it will be equally prompted or suppressed, regardless of the length of the response in which it appears.

### 3.4   Hide and seek: Overlong Reward Shaping

In RL training, we typically set a maximum length for generation, with overlong samples truncated accordingly. We find that improper reward shaping for truncated samples can introduce reward noise and significantly disrupt the training process.

By default, we assign a punitive reward to truncated samples. This approach may introduce noise into the training process, as a sound reasoning process can be penalized solely due to its excessive length. Such penalties can potentially confuse the model regarding the validity of its reasoning process.

To investigate the impact of this reward noise, we first apply an **Overlong Filtering** strategy which masks the loss of truncated samples. We find that this approach significantly stabilizes training and enhances performance, as demonstrated in Figure 5.

Furthermore, we propose **Soft Overlong Punishment** (Equation 13), a length-aware penalty mechanism designed to shape the reward for truncated samples. Specifically, when the response length exceeds the predefined maximum value, we define a punishment interval. Within this interval, the longer the response, the greater the punishment it receives. This penalty is added to the original rule-based correctness reward, thereby signaling to the model to avoid excessively long responses.

$$R_{\text{length}}(y) = \begin{cases} 0, & |y| \leq L_{\max} - L_{\text{cache}} \\ \frac{(L_{\max} - L_{\text{cache}}) - |y|}{L_{\text{cache}}}, & L_{\max} - L_{\text{cache}} < |y| \leq L_{\max} \end{cases} \quad (13)$$

### 3.5   Dataset transformation

Our dataset is sourced from the web and official competition homepages through a combination of web scraping and manual annotation. The answers of math dataset typically come in a variety of formats,

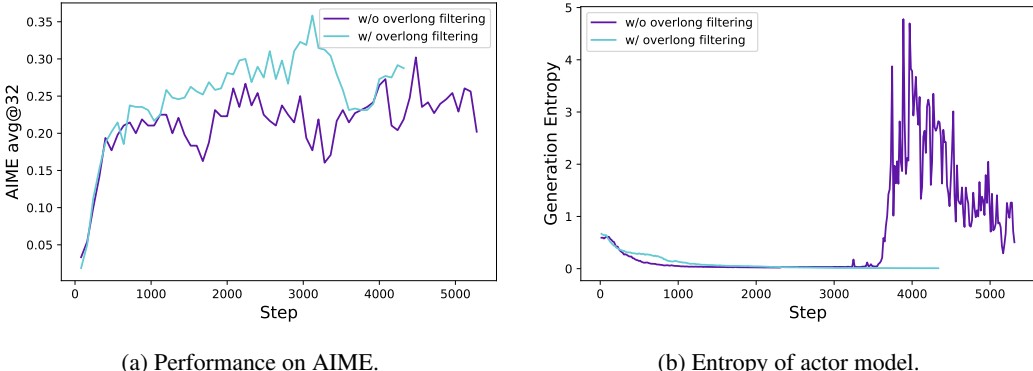

(a) Performance on AIME.         (b) Entropy of actor model.

Figure 5: The accuracy of the actor model on AIME and the entropy of its generation probabilities, both before and after applying **Overlong Reward Shaping** strategy.

---

**Algorithm 1 DAPO**: **D**ecoupled Clip and **D**ynamic s**A**mpling **P**olicy **O**ptimization

---

**Input** initial policy model $\pi_\theta$; reawrd model $R$; task prompts $\mathcal{D}$; hyperparameters $\varepsilon_{\texttt{low}}, \varepsilon_{\texttt{high}}$
1: **for** step = 1,...,M **do**
2:  Sample a batch $\mathcal{D}_b$ from $\mathcal{D}$
3:  Update the old policy model $\pi_{\theta_{old}} \leftarrow \pi_\theta$
4:  Sample $G$ outputs $\{o_i\}_{i=1}^G \sim \pi_{\theta_{\text{old}}}(\cdot|q)$ for each question $q \in \mathcal{D}_b$
5:  Compute rewards $\{r_i\}_{i=1}^G$ for each sampled output $o_i$ by running $R$
6:  Filter out $o_i$ and add the remaining to the dynamic sampling buffer (**Dynamic Sampling** Equation (11))
7:  **if** buffer size $n_b < N$:
8:    **continue**
9:  For each $o_i$ in the buffer, compute $\hat{A}_{i,t}$ for the $t$-th token of $o_i$ (Equation (9))
10:  **for** iteration = 1, ..., $\mu$ **do**
11:    Update the policy model $\pi_\theta$ by maximizing the **DAPO** objective (Equation (8))
**Output** $\pi_\theta$

---

such as expression, formula and number, which makes it challenging to design comprehensive rules to parse them. To provide accurate reward signals using rules and minimize errors introduced by formula parsers, inspired by AIME, we select and transform the answers into integers, which are easy to parse. For example, if the original answer is expressed in the form of $\frac{a+\sqrt{b}}{c}$, we instruct the LLM to modify the question so that the expected answer becomes $a + b + c$. After selection and transformation, we obtained the **DAPO-Math-17K** dataset, which consists of 17K prompts, each paired with an integer as the answer.

## 4 Experiments

### 4.1 Training details

In this work, we focus specifically on mathematical tasks to evaluate our algorithm, which can be readily transferred to other tasks. We adopt the verl framework [20] for training. We use naive GRPO [38] as our baseline algorithm and estimate advantages using group reward normalization.

For hyper-parameters, we utilize the AdamW [39] optimizer with a constant learning rate of $1 \times 10^{-6}$, incorporating a linear warm-up over 20 rollout steps. For rollout, the prompt batch size is 512 and we sample 16 responses for each prompt. For training, the mini-batch size is set to 512, i.e., 16 gradient updates for each rollout step. For **Overlong Reward Shaping**, we set the expected maximum length as 16,384 tokens and allocate additional 4,096 tokens as the soft punish cache. Therefore, the maximum number of tokens for generation is set to 20,480 tokens. As for the **Clip-Higher**

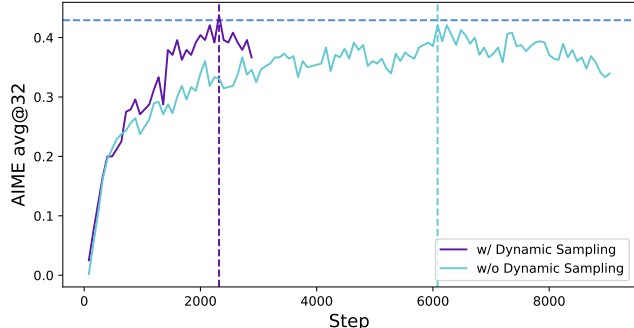

Figure 6: The training progress before and after applying dynamic sampling on a baseline setting.

Table 1: Main results of progressive techniques applied to **DAPO**

| Model | AIME24$_{\text{avg@32}}$ |
|---|:---:|
| **DeepSeek-R1-Zero-Qwen-32B** | 47 |
| Naive GRPO | 30 |
| + Overlong Filtering | 36 |
| + Clip-Higher | 38 |
| + Soft Overlong Punishment | 41 |
| + Token-level Loss | 42 |
| + Dynamic Sampling (**DAPO**) | **50** |

mechanism, we set the clipping parameter $\varepsilon_{\text{low}}$ to 0.2 and $\varepsilon_{\text{high}}$ to 0.28, which effectively balance the trade-off between exploration and exploitation. For evaluation on AIME, we repeat the evaluation set for 32 times and report avg@32 for results stability. The inference hyperparameters of evaluation are set to temperature 1.0 and topp 0.7.

### 4.2 Main results

Experiments on AIME 2024 demonstrate that **DAPO** has successfully trained the Qwen-32B model into a powerful reasoning model, achieving performance superior to DeepSeek's experiments on Qwen2.5-32B using the R1 approach. In Figure 1, we observe a substantial improvement of performance on AIME 2024, with accuracy increasing from near 0% to 50%. Notably, this improvement is achieved with only 50% of the training steps required by DeepSeek-R1-Zero-Qwen-32B.

We analyze the contributions of each training technique in our methodology, as detailed in Table 1. The observed improvements demonstrate the effectiveness of these techniques in RL training, each contributing several accuracy points in AIME 2024. Notably, given the vanilla GRPO setting, only 30% accuracy can be reached by training from a Qwen2.5-32B base model. For token-level loss, although it brings less performance improvement, we find it enhances training stability and makes the length increase more healthily.

When applying **Dynamic Sampling**, although more data needs to be sampled due to the filtering out of zero-gradient data, the overall training time is not significantly affected. As shown in Figure 6, although the number of sampling instances increases, the model's convergence time is even reduced, due to fewer training steps required.

### 4.3 Training dynamics and Case study

Reinforcement learning in large language models is an intrinsically complex system challenge characterized by the interdependence of various subsystems. Modifications to any single subsystem can propagate through the system, leading to unforeseen consequences due to the intricate interplay among these components. Even seemingly minor changes in initial conditions, such as variations in

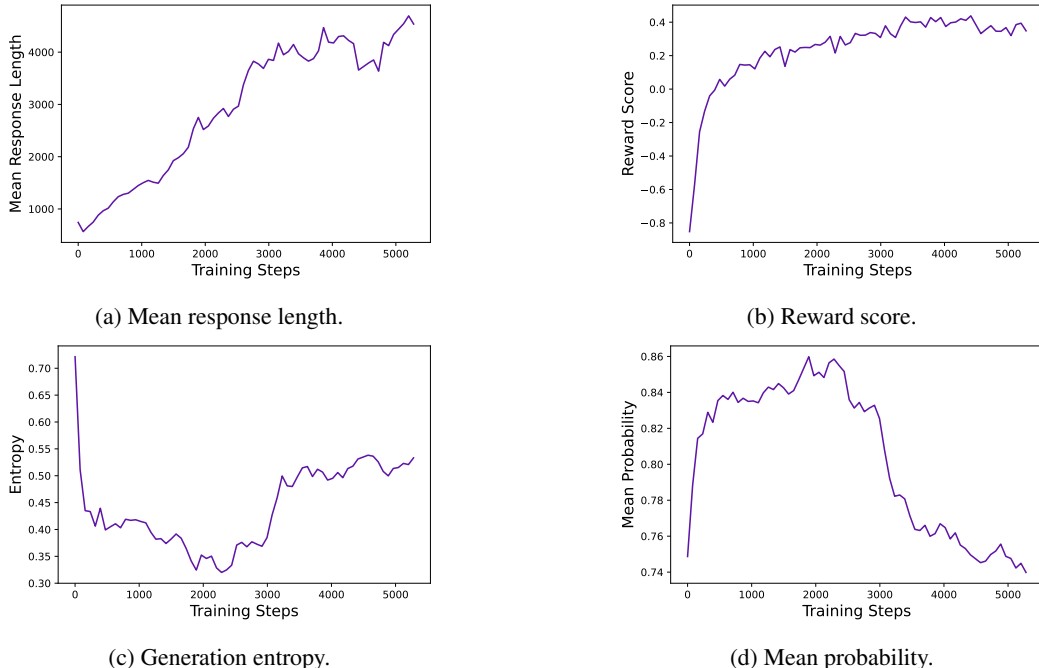

| | |
|---|---|
| (a) Mean response length. | (b) Reward score. |
| (c) Generation entropy. | (d) Mean probability. |

Figure 7: The metric curves of response length, reward score, generation entropy, and the mean probability of **DAPO**, which show the dynamics of RL training and serve as essential monitoring indicators to identify potential issues.

data and hyperparameters, can amplify through iterative reinforcement learning processes, yielding substantial deviations in outcomes. This complexity often confronts researchers with a dilemma: even after meticulous analysis and well-founded expectations that a modification will enhance specific aspects of the training process, the actual results frequently diverge from the anticipated trajectory. Therefore, monitoring of key intermediate results during experimentation is essential for swiftly identifying the sources of discrepancies and, ultimately, for refining the system.

## 4.4 Ablation study

We conduct ablation studies for the key hyperparameter of our proposed Clip-Higher technique, $\varepsilon_{\text{high}}$. The results are shown in Table 2. The default value of the $\varepsilon_{\text{high}}$ is 0.2. We can find that applying Clip-Higher can always achieve a better performance. When the $\varepsilon_{\text{high}}$ is set to 0.28 or 0.3, we can achieve 10 points better than the baseline. The ablation experiments are run for about 3K steps.

Table 2: Validation results of GRPO *w/* Clip-Higher

| $\varepsilon_{\text{high}}$ | 0.2 | 0.25 | 0.28 | 0.3 | 0.4 |
|---|---|---|---|---|---|
| **AIME24**$_{\text{avg@32}}$ | 28.4 | 30.3 | **41.8** | 40.3 | 37.2 |

## 5 Conclusion

In this paper, we release a fully open-sourced system for large-scale LLM RL, including algorithm, code infrastructure, and dataset. The system achieves state-of-the-art large-scale LLM RL performance (AIME 50 using Qwen-32B pretrained model). We propose the **D**ecoupled Clip and **D**ynamic s**A**mpling **P**olicy **O**ptimization (**DAPO**) algorithm, and introduce 4 key techniques to make RL powerfully effective and efficient in the long-CoT RL scenario. Additionally, by open-sourcing the training code and dataset, we provide the broader research community and society with practical access to a scalable reinforcement learning solution, enabling all to benefit from these advancements.

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

# A    Training Dynamics

- **The Length of Generated Responses** is a metric closely related to training stability and performance, as shown in Figure 7a. The increase in length provides the model with a larger space for exploration, allowing more complex reasoning behaviors to be sampled and gradually reinforced through training. However, it is important to note that length does not always maintain a continuous upward trend during training. In some considerable periods, it can exhibit a trend of stagnation or even decline, which has also been demonstrated in [2]. We typically use length in conjunction with validation accuracy as indicators to assess whether an experiment is deteriorating.

- **The Dynamics of Reward** during training has always been one of the crucial monitoring indicators in reinforcement learning, as shown in Figure 7b. In the majority of our experiments, the trend of reward increase is relatively stable and does not fluctuate or decline significantly due to adjustments in experimental settings. This indicates that, given a reliable reward signal, language models can robustly fit the distribution of training set. However, we find that the final reward on the training set often exhibits little correlation with the accuracy on the validation set, which indicates overfitting to the training set.

- **The Entropy of the Actor Model and Generation Probability** are related to the model's exploration capability and are key metrics that we closely monitor in our experiments. Intuitively, the model's entropy needs to be maintained within an appropriate range. An excessively low entropy indicates that the probability distribution is overly sharp, leading to a loss of exploration capability. Conversely, an excessively high entropy is often associated with issues of over-exploration such as gibberish and repetitive generation. For the generation probability, the situation is exactly the opposite. As demonstrated in Section 3.1, by applying the Clip-Higher strategy, we effectively addressed the issue of entropy collapse. In subsequent experiments, we find that maintaining a slow upward trend in entropy is conducive to the improvement of model performance, shown in Figure 7c and Figure 7d.

# B    Dataset Transformation

Here is an example of data transformation, where we demonstrate the original problem, the transformed problem, and the model outputs during the transformation process (the original is in markdown text, which has been rendered for readability).

To ensure the model can complete this task with high accuracy, our goal is to have it perform comprehensive reasoning while avoiding hallucinations. We achieve this by providing a clear thinking framework that encourages thorough reasoning. Specifically, we employ Chain of Thought (CoT) reasoning along with four well-defined steps: extract the answer format, rewrite the problem statement, solve the modified problem and provide an integer as the final answer. For each step, we provide few-shot solutions or detailed guidelines to guide the model's reasoning.

Through this approach, we have observed that in most cases, the LLM can generate reformulations with both format and quality that are satisfactory.

**Original Problem**

Let $x$ and $y$ be real numbers such that $x^2 + y^2 - 22x - 16y + 113 = 0$. Determine the smallest possible value of $x$. **Answer:** $11 - 2\sqrt{6}$

**Transformed Problem**

Let $x$ and $y$ be real numbers such that $x^2 + y^2 - 22x - 16y + 113 = 0$. Determine the smallest possible value of $x$. The original answer is in the form $k - m\sqrt{n}$, where $k$, $m$, and $n$ are integers. Please find the value of $k + m + n$. **Answer:** 19

**Model's output (Markdown Text)**

**Transformation Thought Process**

The original answer to the problem is an algebraic expression $11 - 2\sqrt{6}$. We need to transform this into an integer form. First, note that the answer consists of an integer part and a square root term. According to the given rule, we replace the integer and square root components with undetermined coefficients. The original problem uses $x$ and $y$ as variables, so we choose other letters as undetermined coefficients. We express the answer in the form $k - m\sqrt{n}$ and require the student to find $k + m + n$.

**Answer Format**

$k - m\sqrt{n}$

**Transformed Problem**

Let $x$ and $y$ be real numbers such that $x^2 + y^2 - 22x - 16y + 113 = 0$. Determine the smallest possible value of $x$. The original answer is in the form $k - m\sqrt{n}$, where $k$, $m$, and $n$ are integers. Please find the value of $k + m + n$.

**Answer Calculation for the Transformed Problem**
- The original answer is: $11 - 2\sqrt{6}$.

- The required format is: $k - m\sqrt{n}$.

- Matching the coefficients: $k = 11$, $m = 2$, $n = 6$.

- The final result is: $k + m + n = 11 + 2 + 6 = 19$.

**Final Answer**

19

## C  Case study

---

**Question**:
Given that the base $ABC$ of the tetrahedron $S - ABC$ is an equilateral triangle, the projection $H$ of point $A$ on the face $SBC$ is the orthocenter of $\triangle SBC$, the dihedral angle $H - AB - C$ is $30°$, and $SA = 2$, find the volume of the tetrahedron. The answer is in the form $\frac{k}{m}$, so give the value of $k + m$.

---

**Response**:
Let us consider the given problem: We need to determine ...

...
Next, let $S$ have coordinates $(x_0, y_0, z_0)$. Since $SA = 2$, the distance formula gives us $\sqrt{(x_0)^2 + (y_0)^2 + (z_0)^2} = 2$, so we can write $x_0^2 + y_0^2 + z_0^2 = 4$. Now, remember that $H$ is the orthogonal projection of

...
However, wait a moment, let's rethink about the dihedral angle involving planes in a more thoughtful geometric way. Consider the plane $\alpha_1 = ABC$, which lies entirely on the xy coordinate plane (as $z = 0$). Let the plane $\alpha_2 = SBC$. The point $A$ projected perpendicularly to plane $\alpha_2$ lands on $H$. The line $l = AB$ ...

...

---

Table 3: Emergence of Reflective Behavior in Reinforcement Learning

During the RL training process, we observe an interesting phenomenon: the reasoning patterns of the actor model evolve dynamically over time. Specifically, the algorithm not only reinforces existing reasoning patterns that facilitate correct problem-solving but also gradually gives rise to entirely new modes of reasoning that were initially absent. This finding reveals the adaptability and exploration capability of RL algorithms and offers new insights into the learning mechanisms of the model.

For example, in the early stages of model training, there was virtually no occurrence of checking and reflecting on previous reasoning steps. However, as training progresses, the model exhibits distinct behaviors of reflection and backtracking, as shown in Tables 3 and 4. This observation sheds light on further exploration into interpreting the emergence of reasoning abilities during RL, which we leave for future research.

## D  Limitations

This paper tackles the problem of LLM RL to elicit complex reasoning behaviors. However, only single-turn RL is considered, while multi-turn RL is much more powerful and closer to the real-world applications, which is worth exploring. Besides, for math problems, the paper focuses only on problems with numerical answers, while more challenging mathmatical problems like proof questions are not considered. We leave multi-turn RL and proof scenario as future work.

**Question:**
Among the 900 residents of Aimeville, there are 195 who own a diamond ring, 367 who own a set of golf clubs, and 562 who own a garden spade. In addition, each of the 900 residents owns a bag of candy hearts. There are 437 residents who own exactly two of these things, and 234 residents who own exactly three of these things. Find the number of residents of Aimeville who own all four of these things.

**Response**:
Step by Step Solution:
1. Let's define the following variables:
Let $A$ be the set of residents who own a diamond ring.
Let $B$ be the set of residents who own a set of golf clubs.
Let $C$ be the set of residents who own a garden spade.
Let $D$ be the set of residents who own a bag of candy hearts. Since all 900 residents own a bag of candy hearts, $|D| = 900$.
Let $a_i$ denote the number of residents who own exactly $i$ out of the 4 items.
Let's use the principle of inclusion-exclusion for the union of 4 sets $A, B, C, D$.
...
$a_4 = \frac{-219}{-4} = \frac{219}{4} = 54.75$.
Since $a_4$, the number of people owning all four items, must be a whole number, our current approach needs to be reconsidered in terms of the combinatorial counting of people owning different combinations of the items.
Instead of directly using the inclusion-exclusion principle, we can use a different approach involving the number of people owning combinations of the items $A$, $B$, $C$, and $D$.
Let $x_i$ be the number of people owning exactly $i$ item among $A$, $B$, $C$ (excluding the item $D$ which everyone owns).
Then the total number of people owning exactly $k$ items among the $A$, $B$, $C$, and $D$ can be broken down:
...

Table 4: A case of reflective behavior emergence.

