# OpenReview forum: "DAPO: An Open-Source LLM Reinforcement Learning System at Scale"
_NeurIPS.cc/2025/Conference — NeurIPS 2025 poster_

### Official Review · Reviewer_ZAAH · 2025-06-22

**Clarity:** 3
**Significance:** 3
**Originality:** 4
**Rating:** 4
**Confidence:** 3

**Summary:**

This paper proposes several technical tricks to improve GRPO and collects a new dataset for training. These tricks are insightful and effective (in their experiments). The writing is easy to follow. However, their experimental comparisons are not sufficient.

**Questions:**

1, While the authors solely rely on AIME2024 to assess the performance of their method, I believe that a more comprehensive evaluation would benefit from comparisons across additional benchmarks, such as AMC, MATH500, Minerva Math, and OlympiadBench.

2, The authors conduct experiments with only Qwen2.5-32B. How about Qwen2.5 models in other sizes?

3, The authors claim that their method outperforms the previous SOTA DeepSeek-R1-Zero-Qwen-32B using 50% training steps. My concern is that not all steps are equal. The authors do not mention hyper-parameters like dataset size, batch size and gradient accumulative steps. Therefore, such comparison of training steps can be unfair.

4, The authors propose to remove KL divergence in section 2.3. But they do not include the ablation study of removing KL divergence in Table 1.

5, In my understanding, $\epsilon_{high}$ restricts the increase of r when A > 0, while $\epsilon_{low}$ restricts the decrease of r when A < 0. I recommend the authors to add this in third paragraph because it is not so straightforward to understand why the authors only mention the situation of A > 0.

6, I agree with the idea of filtering out prompts with the accuracy equal to 1 and 0. In my opinion, they exclude part of samples from calculating gradients, but these samples are involved in generating completions. Therefore, when comparing training efficiency in Figure 6, I think the authors should also compare the number of samples for which the model generates completions.

7, For token-level policy gradient loss, in Equation 12, the length of each completion $\left| o_i \right |$ might vary among different batches as $\left| o_i \right |$ will increase with the training steps. As a result, the weight of each token will be different among different batches. How about setting $\left| o_i \right |$ to a proper constant?

**Ethical Concerns:**

["NO or VERY MINOR ethics concerns only"]

**Final Justification:**

I am satisfied with the methods proposed in this paper. Although it looks like multiple tricks, I think useful tricks are important in RL area. But I think reviewer fAi3 's concern "with the unexpected low performance of the GRPO-32B baseline" should be paid attention to.

**Limitations:**

Please refer to the questions.

**Paper Formatting Concerns:**

The first page leaves too much blank.

**Quality:**

3

**Strengths And Weaknesses:**

strengths:

1, This paper focuses on GRPO which is a cutting-edge research direction.

2, This paper proposes insightful and effective technical tricks to improve GRPO.

weaknesses:

1, The experimental comparison are insufficient.

2, Some expressions can be improved to enhance clarity. The detailed comments can be found in questions.

---

> ### Author Rebuttal · Authors · 2025-07-30
>
> We thank reviewer ZAAH for the positive comment on our paper, and address all concerns and questions below.
>
> Q1: Lack of Evaluation:
>
> - We further evaluate the 32B model on additional math benchmarks, where DAPO consistently outperforms GRPO. Additionally, we conduct code experiments on LiveCodeBench, demonstrating that DAPO’s effectiveness extends beyond math tasks.
>
> |                  | AIME24 | MATH500 | AMC  | Minerva Math | Olympiad Bench | LiveCodeBench |
> | ---------------- | ------ | ------- | ---- | ------------ | -------------- | ------------- |
> | DAPO Qwen2.5-32B | 52     | 81      | 88   | 34           | 51             | 62            |
> | GRPO Qwen2.5-32B | 30     | 62      | 72   | 17           | 34             | 50            |
>
> - We evaluated DAPO on three additional backbones, two 7B parameter dense models (Qwen2.5-7B and Qwen2.5-Math-7B) and a 200B parameter MoE model with 20B active parameters. DAPO consistently outperforms baselines across all model sizes and architectures.
>
> |        | GRPOQwen2.5-7B | DAPOQwen2.5-7B | GRPOQwen2.5-Math-7B | DAPOQwen2.5-Math-7B | GRPOIn-house 200A20B MOE | DAPOIn-house 200A20B MOE |
> | ------ | -------------- | -------------- | ------------------- | ------------------- | ------------------------ | ------------------------ |
> | AIME24 | 14             | 20             | 28                  | 36                  | 60                       | 80                       |
>
> Q2: Not equal comparison w/ DeepSeek-R1-Zero-Qwen-32B
>
> - Our dataset size and batch size are presented in Section 4.1. We acknowledge that the comparison cannot be fairly apple-to-apple because DeepSeek didn't reveal any training details or recipes, while just know them train for 10000 steps from DeepSeek R1's paper. But our end2end system performance beats DeepSeek-R1-Zero-Qwen-32B.
>
> Q3: KL Divergence
>
> - We run comparison experiments on Qwen2.5-Math-7B for 3000 steps. The results show w/o KL achieves better performance. Besides, the w/ KL setting already saturates, yet the w/o KL setting achieves better performance and the performance has not saturated.
>
> |             | AIME24 |
> | ----------- | ------ |
> | GRPO w/o KL | 28     |
> | GRPO w/ KL  | 26     |
>
> Q4: More clarification about $\epsilon_{high}$ and A > 0
>
> - Thank you for your suggestion! We will edit and update our paper to clarify it.
>
> Q5: Consider the number of samples generated when comparing training efficiency in Figure 6
>
> - That's an insightful question! Below, we provide a comparison of training efficiency in terms of both generated samples and end-to-end wall time:
>   - Number of Generated Samples: The baseline achieves 42 points after 6080 training steps, while the dynamic sampling approach achieves 43.75 points in just 2320 steps. On average, dynamic sampling generates batches twice the size of the baseline per step. Thus it effectively uses fewer total samples (2320 × 2 = 4640 batches vs. 6080 batches) while achieving superior performance.
>   - Wall Time: With 128 GPUs, the baseline takes approximately 4 days to reach 42 points. In contrast, dynamic sampling reaches 43.75 points in about 2 days, significantly improving efficiency.
> - We have another insight to share, about this substantial reduction in wall time. Practically, sample generation is often bottlenecked by long-tail samples, and increasing the number of samples introduces less additional overhead. Our generation engine uses the Orca scheduler [3], which replaces finished samples with new samples dynamically. Thus, during the generation of the longest sample, several samples could have been generated. In our setting with 128 GPUs, increasing the RL batch size threefold (from 512 to 1536) resulted in only a 1.5x increase in generation wall time.
>
> |                     | AIME score | Num. of generated samples                             | Wall Time (128 H100) |
> | ------------------- | ---------- | ----------------------------------------------------- | -------------------- |
> | Baseline            | 42         | 6080 (step) * batch                                   | 2 days               |
> | w/ dynamic sampling | 43.8       | 2320 (step) * 2 (dynamic times) * batch= 4640 * batch | 4 days               |
>
> Q6: Using a constant as the denominator to normalize loss across batches.
>
> - Empirically, our concurrent work Dr. GRPO [1] proposed to use a constant as the loss denominator and they use the number of maximum tokens. We have tried that and it doesn't work, it even hurt the performance.
>
> |        | GRPOQwen2.5-7B | Dr GRPOQwen2.5-7B | DAPOQwen2.5-7B |
> | ------ | -------------- | ----------------- | -------------- |
> | AIME24 | 14             | 16                | 20             |
>
> - Theoretically, this operation is not grounded and arguable. In the original PPO paper [2], the objective (Equation 1 and 6 of [2]) is expectation over time step t (actions or tokens in the context of LLM). Thus, for each minibatch, we should weigh the loss by dividing it by the number of tokens. This is expectation over time step t.
>
> [1] Understanding R1-Zero-Like Training: A Critical Perspective (https://arxiv.org/pdf/2503.20783)
>
> [2] Proximal Policy Optimization Algorithms (https://arxiv.org/pdf/1707.06347)
>
> [3] Orca: A Distributed Serving System for Transformer-Based Generative Models (https://www.usenix.org/conference/osdi22/presentation/yu)

---

> > ### Comment · Reviewer_ZAAH · 2025-08-04
> > **Reply to rebuttal of authors**
> >
> > Thanks for your reply. It seems that you incorrectly switch the wall time in your table. I am satisfied with the authors's reply. I decide to maintain my score.

---

> > > ### Author Response · Authors · 2025-08-05
> > > **Reply**
> > >
> > > We thank the reviewer again for the positive feedback! Yep we made a mistake and incorrectly switched the wall time in the table. Sorry for the confusion induced by this..

---

### Official Review · Reviewer_Q4hn · 2025-06-26

**Clarity:** 3
**Significance:** 3
**Originality:** 2
**Rating:** 5
**Confidence:** 3

**Summary:**

This paper, DAPO, explores the use of Reinforcement Learning (RL) to enhance the reasoning capabilities of Large Language Models (LLMs). The work is motivated by the observation that previous studies often fail to adequately detail the specific methodologies that contribute to their models' success. The primary contribution of DAPO is the introduction of several optimizations to the standard GRPO baseline, which are claimed to yield significant performance improvements.

These optimizations include:

*   **Asymmetric Clipping:** Utilizing separate low and high clipping parameters within the GRPO loss function.
*   **KL Divergence Removal:** Eliminating the KL divergence term from the GRPO loss.
*   **Dynamic Sampling:** During training, filtering out samples within a batch that have already reached a consensus, as these samples do not contribute to the advantage calculation.
*   **Token-wise Loss Aggregation:** Calculating the loss at the token level directly, rather than first aggregating it per sample and then across the batch.
*   **Soft Overlength Penalty:** Excluding hard-truncated samples from the loss calculation and applying a graduated penalty to samples as they approach the maximum length limit.

**Questions:**

- Could you provide more details on the data source? This information is crucial for assessing potential data contamination and for enabling other researchers to fully understand the experimental setup.
- Could you offer a more detailed justification for your experimental design, particularly for the training protocol and ablation studies? For instance, how many experimental runs were conducted to validate the utility of each technique, and what were the specific criteria for terminating a training run?
- Could you support the claim that removing KL divergence helps? I can't find any quantitative result about that in the paper.

**Ethical Concerns:**

["NO or VERY MINOR ethics concerns only"]

**Final Justification:**

I believe the paper addresses an important problem and contributes to democratizing practical techniques for improving LLM RL training by committing to open-source. While the proposed improvements are not particularly novel or exotic, they provide a straightforward way to boost performance. These insights, in practice, can save substantial GPU-hours and thus hold significant value for many practitioners. The authors have also addressed the limitation that the original evaluation relied on only a single dataset. In addition, they have committed to clarifying the compute resources used and have justified the use of a single seed due to computational cost constraints. For all these reasons, I maintain my original score and recommend acceptance of the paper.

**Limitations:**

I think limitations could be better addressed in the paper, see Weaknesses.

**Paper Formatting Concerns:**

No major formatting issues.
There is a typo in Algorithm 1, it's "reward" not "reawrd".

**Quality:**

2

**Strengths And Weaknesses:**

#### **Strengths**

1.  The paper pragmatically tackles an important and challenging problem. It methodically tests intuitive ideas for improving reasoning performance and supports its findings with corresponding ablation studies.
2.  The paper's commitment to making effective RL training techniques for LLMs more accessible is commendable and a significant benefit to the research community.
3.  The choice of AIME, a challenging mathematical reasoning benchmark, strengthens the paper's contributions, especially given the significant performance gains reported.
4.  The use of standard metrics like avg@32, cons@32, and pass@32 ensures comparability with other work in the field. However, the paper would be improved by explicitly defining these metrics for readers less familiar with this area.

#### **Weaknesses**

1.  The evaluation is limited to a single benchmark (AIME). The authors also appear to have tuned hyperparameters directly on this benchmark's evaluation score, which raises concerns about the generalizability of the proposed techniques.
2.  The paper lacks a comprehensive discussion of its limitations. Aside from a brief paragraph in Appendix D, the broader constraints and potential negative results of the approach are not sufficiently addressed.
3.  Given the paper's goal of promoting reproducibility, the omission of details regarding computational resources (e.g., hardware, budget, runtime) is a notable weakness. This is particularly concerning as the authors answered "No" to the corresponding checklist question 8 about compute.
4.  The reliance on a single random seed for most ablations and plots is a significant limitation. While the high cost of these experiments is understandable, the paper should explicitly acknowledge it and weaken the claims accordingly, or provide additional data points.
5.  The authors appear to have misinterpreted Question 16 of the checklist, which asks about the *use* of LLMs in preparing the manuscript, not whether LLMs are the paper's *topic*.

---

> ### Author Rebuttal · Authors · 2025-07-30
>
> We thank reviewer Q4hn for the positive comment on our paper, and address all concerns and questions below.
>
> Q1: More Evaluation and Generality
>
> - We further evaluate the 32B model on additional math benchmarks, where DAPO consistently outperforms GRPO. Additionally, we conduct code experiments on LiveCodeBench, demonstrating that DAPO’s effectiveness extends beyond math tasks.
>
> |                  | AIME24 | MATH500 | AMC  | Minerva Math | Olympiad Bench | LiveCodeBench |
> | ---------------- | ------ | ------- | ---- | ------------ | -------------- | ------------- |
> | DAPO Qwen2.5-32B | 52     | 81      | 88   | 34           | 51             | 62            |
> | GRPO Qwen2.5-32B | 30     | 62      | 72   | 17           | 34             | 50            |
>
> - We evaluated DAPO on three additional backbones, two 7B parameter dense models (Qwen2.5-7B and Qwen2.5-Math-7B) and a 200B parameter MoE model with 20B active parameters. DAPO consistently outperforms baselines across all model sizes and architectures.
>
> |        | GRPOQwen2.5-7B | DAPOQwen2.5-7B | GRPOQwen2.5-Math-7B | DAPOQwen2.5-Math-7B | GRPOIn-house 200A20B MOE | DAPOIn-house 200A20B MOE |
> | ------ | -------------- | -------------- | ------------------- | ------------------- | ------------------------ | ------------------------ |
> | AIME24 | 14             | 20             | 28                  | 36                  | 60                       | 80                       |
>
> Q2: Details regarding computational resources
>
> - The computation resource cost is 128 H100 GPUs for one week.
>
> Q3: More discussion of limitations & A single random seed is a limitation.
>
> - Thank you for kindly raising this point. We will expand our discussion on limitations as follows:
>   - The outcome reward currently provides minimal guidance on the reasoning process, resulting in inefficient and prolonged reasoning chains. Future research could explore more detailed process-based reward designs to potentially shorten reasoning length.
>   - The main experiments were conducted using only a single random seed due to the substantial computational resources required for large-scale LLM RL experiments. We acknowledge this as an important limitation.
>   - Regarding safety, the existing verifier-based RL focuses solely on answer correctness without supervising model output safety. Future work could address this by incorporating both outcome verification and a general safety reward model.
>
> Q4: Misinterpretation of checklist question 16:
>
> - Thank you for pointing out this oversight :). We will correct it and notice it next time.
>
> Q5: Could you provide more details on the data source?
>
> - We crawl all questions from the AOPS (https://artofproblemsolving.com/) website and conducted a thorough decontamination of evaluation sets.
>
> Q6: How many experimental runs were conducted to validate the utility of each technique, and what were the specific criteria for terminating a training run?
>
> - Given the high computational cost, we performed only one run per technique for validation. Training runs were terminated upon reaching performance saturation.
>
> Q7: Could you support the claim that removing KL divergence helps?
>
> We run comparison experiments on Qwen2.5-Math-7B for 3000 steps. The results show w/o KL achieves better performance. Besides, the w/ KL setting already saturates, yet the w/o KL setting achieves better performance and the performance has not saturated.
>
> |             | AIME24 |
> | ----------- | ------ |
> | GRPO w/o KL | 28     |
> | GRPO w/ KL  | 26     |

---

> > ### Comment · Reviewer_Q4hn · 2025-08-04
> >
> > I thank the authors for their rebuttal, I think the additional results and explanations strengthen the claims made in the paper. I therefore maintain my rating.

---

> > > ### Author Response · Authors · 2025-08-05
> > > **Reply**
> > >
> > > We thank the reviewer again for the positive feedback!

---

### Official Review · Reviewer_fAi3 · 2025-07-03

**Clarity:** 2
**Significance:** 1
**Originality:** 2
**Rating:** 2
**Confidence:** 5

**Summary:**

The paper introduces DAPO, an open-source RL training techniques in LRMs. The authors propose four key techniques: Clip-Higher, Dynamic Sampling, Token-Level Policy Gradient Loss, and Overlong Reward Shaping. Their implementation achieves impressive results on the AIME 2024 benchmark, scoring 50 points with the Qwen2.5-32B base model.

**Questions:**

- Why doesn't Figure 4 correlate entropy collapse with model performance metrics? This connection would provide critical insight into how entropy dynamics affect reasoning capabilities.
- The paper emphasizes the epsilon lower bound's importance in Clip-Higher but provides no experimental analysis of its effects.
- Additionally, in Figure 6, the baseline and proposed method are trained for different numbers of steps, with the baseline appearing nearly converged - how does this represent a fair comparison of both methods at their optimal performance points?

**Ethical Concerns:**

["NO or VERY MINOR ethics concerns only"]

**Final Justification:**

Based on the current description, the contribution on the methodology side mainly comes from the dynamic filtering and the clip-higher strategy. Consequently, I maintain that **the novelty is not sufficiently strong in its current form**, despite the soundness of combining many tricks to boost the GRPO performance. Currently, I maintain my current rating as the reasons to reject outweigh the reasons to accept, with **the main limitations being the strength of the novelty claim and the credibility of empirical evidence**.

**Limitations:**

Yes

**Quality:**

2

**Strengths And Weaknesses:**

### Strengths:
- The authors provide fully open-source code with straightforward, easily reproducible techniques.
- The paper is well-written and easy to understand.

### Weaknesses
- The evaluation relies primarily on AIME 2024 test set, which contains only 30 problems. This narrow benchmark scope creates significant uncertainty about whether the method's success is genuinely robust or merely coincidental. Testing on a single model backbone (Qwen2.5-32B) further compounds this concern about reproducibility across different model architectures.
- The four proposed techniques appear to be incremental adjustments rather than fundamental methodological advances:
1) The Clip-Higher approach is essentially parameter tuning of existing clipping mechanisms
2) The Dynamic Sampling method addresses a known issue (all-zero or all-one samples providing little training signal) but doesn't solve the underlying exploration challenge
3) Both the Token-Level Policy Gradient Loss and Overlong Reward Shaping techniques are essentially simple penalties or balancing mechanisms for sentence length rather than novel approaches to the core reinforcement learning problems
- The paper fails to provide comprehensive information about the computational resources required for training. This omission significantly hinders reproducibility efforts, as other researchers cannot accurately assess the hardware requirements or computational budget necessary to implement this approach.

---

> ### Author Rebuttal · Authors · 2025-07-30
>
> We thank reviewer fAi3 for reviewing our paper, and address all concerns and questions below.
>
> Q1: Evaluation of more datasets and model backbones.
>
> - We further evaluate the 32B model on additional math benchmarks, where DAPO consistently outperforms GRPO. Additionally, we conduct code experiments on LiveCodeBench, demonstrating that DAPO’s effectiveness extends beyond math tasks.
>
> |                  | AIME24 | MATH500 | AMC  | Minerva Math | Olympiad Bench | LiveCodeBench |
> | ---------------- | ------ | ------- | ---- | ------------ | -------------- | ------------- |
> | DAPO Qwen2.5-32B | 52     | 81      | 88   | 34           | 51             | 62            |
> | GRPO Qwen2.5-32B | 30     | 62      | 72   | 17           | 34             | 50            |
>
> - We evaluated DAPO on three additional backbones, two 7B parameter dense models (Qwen2.5-7B and Qwen2.5-Math-7B) and a 200B parameter MoE model with 20B active parameters. DAPO consistently outperforms baselines across all model sizes and architectures.
>
> |        | GRPOQwen2.5-7B | DAPOQwen2.5-7B | GRPOQwen2.5-Math-7B | DAPOQwen2.5-Math-7B | GRPOIn-house 200A20B MOE | DAPOIn-house 200A20B MOE |
> | ------ | -------------- | -------------- | ------------------- | ------------------- | ------------------------ | ------------------------ |
> | AIME24 | 14             | 20             | 28                  | 36                  | 60                       | 80                       |
>
> Q2: Novelty & Clip-higher is hyperparameter tuning
>
> - **Effective tricks are important to make RL actually work, especially in large-scale** **LLM** **training.** Our methods achieve a significant improvement of 20 points on AIME (the core benchmark adopted by OpenAI-O1, DeepSeek R1) and surpass DeepSeek-R1's RL. To date, few other GRPO-based methods can surpass DeepSeek-R1's performance purely using RL in the zero setting. We achieved this, and the effective algorithm details are the keys to making this happen.
> - **Achieving top-tier performance with a simple method is inherently a kind of innovation.** Our conceptually simple method surpasses DeepSeek-level RL performance. Few open research have reached DeepSeek-level RL performance. Not only have we accomplished this, but we have also made all our detailed recipes, code, and data fully open-source, promoting transparency and reproducibility.
> - Clip-Higher: it goes beyond "hyperparameter-tuning" and encodes crucial algorithm insights.
>   - "Clip-Higher" is a novel approach not previously well-known or utilized by the broader research community. Specifically, few people know that **Clip-Higher can effectively balance exploration and exploitation**. Besides, Clip-Higher’s approach to independently setting upper and lower clip bounds is novel, where traditionally both clip bounds are identical. Despite its simplicity, it demonstrates exceptional effectiveness. We anticipate that our clear demonstration of its efficacy will inspire its adoption and broader use within the research community.
>   - Pure "hyperparameter-tuning" has a vast search space and cannot lead to Clip-Higher. Our identification of Clip-Higher is a targeted, insight-driven adjustment rather than random exploration or just "tuning". RL systems contain hundreds of hyperparameters that can be tuned. To name a few, just for the entropy problem, temperature, temperature scheduler, entropy bonus weights, entropy loss weights, kl loss weights (kl helps entropy), epsilon-greedy sampling ratios, top-k/top-p threshold... These are just "hyperparameter tuning". Clip-Higher is different from these simple tuning and emerges as both novel and remarkably effective, making it an elegant and powerful solution.
> - Additionally, a major contribution of our work is clearly **identifying critical issues in applying RL to** **LLMs** and providing **practical solutions to make the successful application of RL possible**. Our focused analysis, e.g. about entropy, will likely inspire further research in this important area.
>
> Q3: Dynamic sampling addresses a known issue but doesn't solve the underlying exploration challenge.
>
> - We argue that few LLM reasoning RL papers pointed this out or addressed this problem. We are the first to provide an effective solution to address this problem, which is shown to speed up training and improve performance.
> - The exploration challenge is addressed by Clip-higher. Clip-higher presents a clearly effective solution to this challenge. (shown in Figure 2, it raises entropy and performance). Dynamic sampling is not designed to address the exploration challenge.
>
> Q4: Token-Level PG Loss and Overlong Reward Shaping are not novel approaches to the core RL problems
>
> - We acknowledge that these two techniques are not novel. However, the goal of our paper is not to propose an "entirely new method for the core RL". Instead, we aim to demonstrate how RL can be successfully applied to significantly enhance LLM reasoning practically, surpassing state-of-the-art DeepSeek-R1 level RL performance. Through our experiments, we identify them as crucial components for achieving this level of performance and transparently share these findings with the research community. Note that the community continues to face challenges in replicating DeepSeek-R1-level RL results, underscoring the practical value of our contributions.
>
> Q5: Information of computational resources required for training
>
> - The computation resource cost is 128 H100 GPUs for one week.
>
> Q6: Correlation between entropy collapse and model performance:
>
> - We show the correlation in Figure 2. We can clearly see that for the baseline trial, the entropy collapses to its minimum at around 750 steps, and the peak performance is achieved also at round 750 steps.
>
> Q7: Epsilon lower bound's importance is emphasized but no experimental analysis
>
> - Our proposed method is Clip-Higher and is about the epsilon upper bound. The epsilon lower bound is not our focus and we just keep it as it is.
>
> Q8: In Figure 6, the proposed method and baseline are trained for different number of steps, where the baseline is converged.
>
> - Our method surpasses the converged baseline using significantly fewer training steps, clearly demonstrating its superior efficiency and performance.

---

> ### Comment · Reviewer_fAi3 · 2025-08-05
>
> Thank the authors for the response. After carefully considering the authors' rebuttal, I remain unconvinced and have summarized below the specific points I still find unclear. If I’ve misunderstood anything or any of my objections are misplaced, I would greatly appreciate your clarification. The remaining concerns need to be addressed as follows.
> - The authors present dynamic filtering as a contribution, but this is essentially identical to the online filtering approach already proposed in Prime[1].
> - Despite the authors' defense, the Clip-Higher approach appears to be primarily a hyperparameter adjustment rather than a methodological innovation. The authors need to better articulate why this isn't simply parameter tuning.
> - The reported GRPO baseline with 32B models performs significantly worse than expected. For example, your baseline scores 62 on MATH500 and 30 on Olympiad, while Qwen-2.5-7B-base models using GRPO in the VERL[2] framework in OpenR1-Math-220K dataset[3] achieved 80 on MATH500 and 41 on Olympiad with just 7B models. This suggests implementation issues that may artificially inflate the proposed method's improvements.
> - Results focus too heavily on AIME. Please provide comprehensive results across all benchmarks for each model configuration, including average performance metrics.
> - Based on the reward curves, the experiments appear to have converged early. Why not terminate training earlier and run multiple trials to obtain more statistically meaningful results?
>
>
> These issues need to be adequately addressed to validate the paper's claims and contributions.
>
> [1] Process reinforcement through implicit rewards.
>
> [2] https://github.com/volcengine/verl
>
> [3] https://huggingface.co/datasets/open-r1/OpenR1-Math-220k

---

> ### Author Response · Authors · 2025-08-07
>
> Q1:
> - The two techniques are fundamentally different. Online filtering doesn't improve performance because they "filter" samples, causing the effective batch size to shrink during training—this destabilizes gradients and brings no clear performance benefit. However, in `dynamic sampling`, the batch size is constant and yields steadier gradient estimates and more reliable learning. This is a significant difference and technical improvement. And performance improvement shows that.
> - Prime is an arxiv paper and actually our contemporary work. We develop this method independently.
>
> Q2:
> - Nobody has previously shown that this hyperparameter tuning can balance exploration and exploitation, an insight that represents an innovation.
> - No prior work has shown that this tuning delivers such a substantial performance gain, highlighting Clip-Higher as the true innovation.
>
> Q3:
> - The dataset is significantly different and the comparison does't hold. They use a 220K dataset while we use a 17K dataset.
> - Our DAPO-32B results beat DeepSeek's RL results. This is a fact and undebateble. We will release the checkpoint for reproduction.
>
> Q4:
> - We provide more performance evaluation across different datasets and model architectures in Q1 of the initial rebuttal. We think this review just ignores the key and obvious facts presented in our initial rebuttal.
>
> Q5:
> - The evaluation score keeps growing (Figure 1) and the reward score almost keeps growing (Figure 7 b). No reason to terminate the training earlier. We think this comment just ignores the key and obvious facts presented in our paper.
>
> Thank you for taking the time to review our work. However, several of your comments appear to disregard key evidence and facts presented in the paper and in our initial rebuttal, which has hindered productive dialogue. At the same time, all other reviewers have at least read through the contents of our paper, and offered positive feedback. We will discuss this concern with the Area Chair to identify the potentially `irresponsible review`.

---

> > ### Comment · Reviewer_fAi3 · 2025-08-08
> >
> > `Q1:`
> >
> > - Prime's arXiv v1 version[1] was published on February 3, 2025, so it's not really contemporaneous work with DAPO.
> > - Additionally, dynamic sampling and online filtering are fundamentally similar in their filtering approach. Based on DAPO's ablation study results, **dynamic sampling doesn't actually demonstrate performance improvements**, and the observed differences could very likely be attributed to the inherent fluctuations in RL algorithms.
> >
> > | Method |  maj@32 | best@32 | mean@32 |
> > | --- | --- | --- | --- |
> > | dapo | 63.07 | 79.66 | 52.08 |
> > | dapo w/o dynamic sampling | 63.08 | 82.24 | 50.41 |
> >
> > ***
> >
> > `Q2:`
> >
> > In RL algorithms like PPO or GRPO, clipping is specifically designed to prevent large divergence between old and new policies. Increasing the clip parameter encourages exploration, which is common knowledge in RL domains. This innovation point is insufficient, and the parameters DAPO provides aren't generalizable enough to other scenarios.
> >
> > ***
> >
> > `Q3:`
> >
> > Our concern is that the baseline results seem unusually low. We're not comparing with non-aligned experiments, but most models [2,3] (even smaller-sized models [4]) trained with GRPO achieve above 75 on Math500. Notably, Qwen2.5-32B's official starting point is already approaching your GRPO fine-tuning results. This is especially concerning when even smaller-sized models trained with GRPO achieve significantly better performance. These discrepancies raise doubt on the validity of DAPO's dramatic comparative improvements.
> >
> > ***
> >
> > `Q4:`
> >
> > In the initial rebuttal, the result in Q1 only included the implementations using Qwen2.5-32B. Our comment was to provide more comprehensive results across different model configurations. Given the very limited time remaining before the rebuttal deadline, we believe this is no longer a factor that affects the overall evaluation of the proposed method.
> >
> > ***
> >
> > `Q5:`
> >
> > RL algorithms inherently have fluctuations, which is why multiple seeds should be run for reliable patterns. Given DAPO's computational costs, we suggested earlier termination to allow for multiple seeds. If you have resources for extended training on multiple seeds and steps, we'd be interested in those results.
> >
> > [1] Process reinforcement through implicit rewards. [v1] Mon, 3 Feb 2025 15:43:48 UTC
> >
> > [2] https://github.com/TsinghuaC3I/Awesome-RL-Reasoning-Recipes
> >
> > [3] https://huggingface.co/internlm/OREAL-32B
> >
> > [4] https://huggingface.co/agentica-org/DeepScaleR-1.5B-Preview
> >
> > [5] https://qwenlm.github.io/blog/qwen2.5-llm/
> >
> > ***
> >
> > We thank the authors for their response in the last round. However, the raw response hardly addresses my concerns. Particularly, the authors deliberately mislead our concerns as an irresponsible review due to two reasons that don't hold true: 1) we disregard key evidence and facts presented in the paper and initial rebuttal, and 2) all other reviewers offered positive feedback.
> >
> > First, we do not disregard any key evidence and facts as
> >
> > - The Q1 is concerned with the novelty of dynamic filtering, given that a very similar approach has been proposed by an earlier work posted on arXiv three months earlier than the NeurIPS 2025 deadline. Also, the ablation on dynamic filtering does not verify its claim.
> > - The Q2 is concerned with the novelty of the clip-higher strategy, given that the clip-higher strategy in PPO is common knowledge for promoting exploration in RL domains.
> > - The Q3 is concerned with the unexpected low performance of the GRPO-32B baseline, which raises doubt on the validity of DAPO's dramatic comparative improvements.
> > - The Q4 is concerned with the comprehensiveness of evaluation across different model configurations.
> > - The Q5 is concerned with the suggestion of running multiple trials to obtain more statistically meaningful results.
> >
> > Second, even if other reviewers offered positive comments, **I still reserve the right to retain my own opinions**. I have responded as promptly as possible to discuss any points of disagreement, offered pros and cons of the paper, and clearly expressed my concerns to provide high-quality review comments.
> >
> > We argue that, **misleading our concerns as an irresponsible review before fully addressing them in previous rounds is indeed irresponsible for the author-reviewer discussion process, as well as disrespectful to the reviewers**. This violates the code of conduct of NeurIPS.
> >
> > Based on the current description, the contribution on the methodology side mainly comes from the dynamic filtering and the clip-higher strategy. Consequently, I maintain that the **novelty is not sufficiently strong** in its current form, despite the soundness of combining many tricks to boost the GRPO performance. Currently, I maintain my current rating as the reasons to reject outweigh the reasons to accept, with **the main limitations being the strength of the novelty claim and the credibility of empirical evidence.**

---

> > > ### Comment · Reviewer_ZAAH · 2025-08-08
> > >
> > > I agree with reviewer fAi3's opinion that "even if other reviewers offered positive comments, I still reserve the right to retain my own opinions". I give my comments about this paper based on my knowledge, which does not mean that my knowledge can cover all aspects of this paper (although I am trying to). fAi3 does reserve the right to express his/her opinions about this paper, which should not be influenced by my opinions.

---

### Official Review · Reviewer_ADmi · 2025-07-04

**Clarity:** 4
**Significance:** 3
**Originality:** 2
**Rating:** 5
**Confidence:** 4

**Summary:**

This paper dives into the recipe of large-scale RLHF for LLMs. The authors point out that while many works such as o1 or DeepSeek R-1 do include their RLHF methods in their reports, they all hold out many details, preventing the open source LLM community from reproducing their results. Hence, the authors introduce DAPO, an RL algorithm designed to overcome common issues like instability and entropy collapse in training large language models for complex reasoning tasks. The authors present four novel techniques, i.e., Clip-Higher, Dynamic Sampling, Token-Level Policy Gradient Loss, and Overlong Reward Shaping, that collectively enhance training efficiency and performance. By applying DAPO to a Qwen2.5-32B model, they achieve a state-of-the-art 50% accuracy on the AIME 2024 math benchmark, outperforming prior work in fewer training steps. The authors also commit to the full open-sourcing of the algorithm, training code, and a new 17K-prompt math dataset to improve reproducibility and foster further research in the community.

**Questions:**

1. For methods except for Clip-Higher, they are generalizable since they are more like engineering improvements than hyperparameter tuning. For Clip-Higher, it looks like this is pure parameter tuning to balance exploration and exploitation on math tasks. It's not clear if it works well in general.

2. In Sec 3.3, i.e., Token-Level Policy Gradient Loss, the authors mentioned that long samples can be either of high-quality, or of "low-quality patterns such as gibberish and repetitive words". How does the proposed token-level sampling, which effectively increases long samples' loss weight as a blanket policy, tackle both scenarios at the same time?

3. In Sec 3.4, i.e., Overlong Reward Shaping, I don't quite get why we should by default "assign a punitive reward to truncated samples" (L151), which causes this problem in the first place. As far as I can tell, RL methods having a maximum length doesn't necessary mean that samples got truncated due to excessive length gets extra punishiment as long as the discounted reward computation traces back the whole sequence. Also, if one desires longer sequences, wouldn't increase maximum length just solve the problem?

**Ethical Concerns:**

["NO or VERY MINOR ethics concerns only"]

**Final Justification:**

Thanks the authors for additional remarks. I am satisfied that the evaluation of the proposed method is no longer an issue. Hence, I decide to raise my rating.

**Limitations:**

I would argue that the paper lacks novelty in general. The proposed methods sound more like a better implementation of GRPO rather than a new RL method. However, I do agree that a good implemtation is very important and often very critical in RL.

**Paper Formatting Concerns:**

Eq (8), (10), (11) and (12) are very repetitive. Consider merging them together.

**Quality:**

4

**Strengths And Weaknesses:**

Strengths:
* All techniques developed by the authors are clearly explained with a good motivation.
* The DAPO system achieves a 50% accuracy on the AIME 2024, surpassing naive GRPO by 20 points with only half amout of training.
* Ablation studies are well presented to justify the proposed methods.

Weaknesses:
* Lack of novelty. The paper sounds more like a good tech report showing off parameter tuning and smart engineering efforts rather than an algorithmic contribution.
* Lack of evaluation on other forms of tasks. It's not clear if the proposed methods would still work for other domains such as code generation or standard VQAs.

---

> ### Author Rebuttal · Authors · 2025-07-30
>
> We thank reviewer ADmi for the positive comment on our paper, and address all concerns and questions below.
>
> Q1: Lack of novelty
>
> - **Effective tricks are important to make RL actually work, especially in large-scale** **LLM** **training.** Our methods achieve a significant improvement of 20 points on AIME (the core benchmark adopted by OpenAI-O1, DeepSeek R1) and surpass DeepSeek-R1's RL. To date, few other GRPO-based methods can surpass DeepSeek-R1's performance purely using RL in the zero setting. We achieved this, and the effective algorithm details are the keys to making this happen.
> - **Achieving top-tier performance with a simple method is inherently a kind of innovation.** Our conceptually simple method surpasses DeepSeek-level RL performance. Few open research have reached DeepSeek-level RL performance. Not only have we accomplished this, but we have also made all our detailed recipes, code, and data fully open-source, promoting transparency and reproducibility.
> - Clip-Higher: it goes beyond "hyperparameter-tuning" and encodes crucial algorithm insights.
>   - "Clip-Higher" is a novel approach not previously well-known or utilized by the broader research community. Specifically, few people know that **Clip-Higher can effectively balance exploration and exploitation**. Besides, Clip-Higher’s approach to independently setting upper and lower clip bounds is novel, where traditionally both clip bounds are identical. Despite its simplicity, it demonstrates exceptional effectiveness. We anticipate that our clear demonstration of its efficacy will inspire its adoption and broader use within the research community.
>   - Pure "hyperparameter-tuning" has a vast search space and cannot lead to Clip-Higher. Our identification of Clip-Higher is a targeted, insight-driven adjustment rather than random exploration or just "tuning". RL systems contain hundreds of hyperparameters that can be tuned. To name a few, just for the entropy problem, temperature, temperature scheduler, entropy bonus weights, entropy loss weights, kl loss weights (kl helps entropy), epsilon-greedy sampling ratios, top-k/top-p threshold... These are just "hyperparameter tuning". Clip-Higher is different from these simple tuning and emerges as both novel and remarkably effective, making it an elegant and powerful solution.
> - Additionally, a major contribution of our work is clearly **identifying critical issues in applying RL to** **LLMs** and providing **practical solutions to make the successful application of RL possible**. Our focused analysis, e.g. about entropy, will likely inspire further research in this important area.
>
> Q2: Lack of evaluation and generality
>
> - We further evaluate the 32B model on additional math benchmarks, where DAPO consistently outperforms GRPO. Additionally, we conduct code experiments on LiveCodeBench, demonstrating that DAPO’s effectiveness extends beyond math tasks.
>
> |                  | AIME24 | MATH500 | AMC  | Minerva Math | Olympiad Bench | LiveCodeBench |
> | ---------------- | ------ | ------- | ---- | ------------ | -------------- | ------------- |
> | DAPO Qwen2.5-32B | 52     | 81      | 88   | 34           | 51             | 62            |
> | GRPO Qwen2.5-32B | 30     | 62      | 72   | 17           | 34             | 50            |
>
> - We evaluated DAPO on three additional backbones, two 7B parameter dense models (Qwen2.5-7B and Qwen2.5-Math-7B) and a 200B parameter MoE model with 20B active parameters. DAPO consistently outperforms baselines across all model sizes and architectures.
>
> |        | GRPOQwen2.5-7B | DAPOQwen2.5-7B | GRPOQwen2.5-Math-7B | DAPOQwen2.5-Math-7B | GRPOIn-house 200A20B MOE | DAPOIn-house 200A20B MOE |
> | ------ | -------------- | -------------- | ------------------- | ------------------- | ------------------------ | ------------------------ |
> | AIME24 | 14             | 20             | 28                  | 36                  | 60                       | 80                       |
>
> Q3: Clip-Higher is purely hyperparameter tuning on math tasks and perhaps not generalizable.
>
> - Our analysis is about the entropy dynamics of the RL process and no specific data type assumption is made. Principlely, this is a general RL trick and can generalize. We conduct a quick experiment and show that on coding tasks, clip-higher is also effective. Besides, we argue that clip-higher together with its superior performance encode algorithmic insights and the related details can be found in Q1.
>
> |                                    | LiveCodeBench |
> | ---------------------------------- | ------------- |
> | DAPO Qwen2.5-32B (only Clip-Higer) | 56            |
> | GRPO Qwen2.5-32B                   | 50            |
>
> Q4: How token-level loss tackle both scenarios.
>
> - The RL training objective has two types of effects: it either encourages (increases likelihood) or punishes (decreases likelihood). Positive samples are encouraged, while negative samples are punished. When applying token-level loss, longer samples naturally receive greater weight; thus, positive samples are encouraged more strongly, and negative samples face increased punishment, significantly reducing their likelihood. This means token-level loss can naturally tackle both scenarios.
>
> Q5: Why is a punitive overlong reward by default? If one desires longer sequences, wouldn't increase maximum length just solve the problem?
>
> - Regarding your second point, autoregressive LLM generation is typically the primary time bottleneck of the whole system. Therefore, in practice, it is necessary to impose a maximum generation length during the rollout phase to ensure efficiency.
> - The default punitive strategy for overly long responses is common practice in verifier-based LLM RL, exemplified by DeepSeek R1. Such methods use accuracy rewards, giving positive feedback only if the response is correct and is validated via robust, rule-based verification (see Sec 2.2.2 of DeepSeek R1). Overlong samples cannot pass such rule-based verification, because they cannot give a correct answer.

---

### Decision · Program_Chairs · 2025-09-17

**Decision:**

Accept (poster)

**Comment:**

The paper introduces DAPO, a practical recipe for large-scale RLHF in LLMs. It proposes four techniques—Clip-Higher, Dynamic Sampling, Token-Level PG Loss, and Overlong Reward Shaping—to improve GRPO training stability and efficiency. Applied to Qwen2.5-32B, DAPO achieves higher accuracy on AIME 2024, surpassing prior work. The code, data and models are all open-source.

This paper has clear exposition of practical RL training improvements, strong empirical results on AIME and additional math/coding benchmarks and significant efficiency gains (fewer steps, better stability). Full open-sourcing commitment enhances reproducibility and community impact.

However, the novelty is somewhat limited, as most techniques resemble parameter tuning or known adjustments. Baseline GRPO results appear weaker than expected, raising questions on fairness of comparisons. Evaluation is heavily math-centric, broader tasks and multiple seeds are missing. And the compute costs and limitations initially underreported. The reviewers raised or maintained the score during the rebuttal.

Despite limited novelty, DAPO makes a useful, reproducible, and empirically strong contribution to RL for LLM reasoning. It can be accepted as a practical paper.